# AUTOREGRESSIVE GRAPH NETWORK FOR LEARNING MULTI-STEP PHYSICS

## ABSTRACT

In this work, we propose a *Autoregressive* Graph Network (AGN) that learns forward physics using a temporal inductive bias. Currently, temporal state space information is provided as additional input to a GN when generating roll-out physics simulations. While this relatively increases the network's predictive performance over multiple time steps, a temporal model enables the network to induce and learn temporal biases. In dynamical systems, the arrow of time simplifies possible interactions in the sense that we can assume current observations to be dependent on preceding states. Our proposed GN encodes temporal state information using an autoregressive encoder that can parallelly compute latent temporal embeddings over multiple time steps during a single forward pass. We perform case studies that compare multi-step forward predictions against baseline data-driven one-step GNs as well as multi-step sequential models across diverse datasets that feature different particle interactions. Our approach outperforms the baseline GN and physics-induced GNs in 8 out of 10 and in 8 out of 10 particle physics datasets respectively when conditioned on optimal historical states. Further, through an energy analysis we find that our method not only accumulates the least roll-out error but also conserves energy more efficiently than baseline Graph Transformer Network while having an order of magnitude lesser parameters.

## 1 INTRODUCTION

In the recent years, there has been a growing interest in learning physics with the help of deep learning coupled with other techniques such as inductive biases, physics informed loss functions and meta-learning (Fragkiadaki et al. (2016); Battaglia et al. (2016); Xu et al. (2019); Hall et al. (2021)). Relational networks such as Graph Networks (GNs) can decompose and learn the dynamics of a physics system on the basis of particle interactions within their neighborhoods (Battaglia et al. (2016); Li et al. (2018); Sanchez-Gonzalez et al. (2020)). Across science and engineering, particle states often contain system and particle specific properties such as mass, density, velocity, particle type, etc. that are required to approximate the dynamics of a system.

In general, given the current state of a system of particles along with particle specific local properties and global system properties, it is possible to apply GNs to predict the trajectory of the system (Sanchez-Gonzalez et al. (2018; 2020)). Often referred to as the forward problem, it assumes knowledge about the physical properties of particles and therefore utilizes the observations to construct a suitable model that predicts the trajectory of the system of particles. The solution to a typical forward dynamics problem governed by an ODE involving particles can be parameterized using a GNN by learning from the current state or by using a history of previous particle states. There are strong benefits to training on entire sequences or multiple time-steps (Mohajerin (2017); Xu et al. (2019)) as one-step GNs tend to be unstable and accumulate error in the long-run. While prior work has shown that concatenating history of previous states enables a trained simulator such as a GNN to predict the next state more accurately, a sequential model captures certain symmetries, e.g., arrow of time, conservation of energy, momentum .etc. Sequential models such as RNN, LSTM, GRU and Transformers have been applied to 1D time series and $N$-body systems (Chen et al. (2018); Zhang et al. (2020); Han et al. (2022)). While appealing choices to model dynamical systems due to their implicit memory mechanisms, they require sequential computations that come with significant memory overhead as the lookback length and/or the dimensionality of the problem increases.

To avoiding sequential computations while retaining relational information about previous states, masking has been employed as a successful strategy in enabling feed-forward neural networks to enjoy the best of both worlds (Germain et al. (2015); Van Den Oord et al. (2016); Papamakarios et al. (2017)). Autoregressive models have been widely used across Machine Learning to learn conditional dependencies to model distributions Rezende & Mohamed (2015); Papamakarios et al. (2017) and learn long range predictions using masked/causal Transformer like models (Ghazvininejad et al. (2019); Shi et al. (2020); Han et al. (2022). Further, to improve predictive performance, the original feature space is often mapped to a hidden latent space on which Transformers and the family of sequential models operate. The map from the orginal feature space to a latent space at each instant in time is often performed using linear layers such as a MLP. Much of the previous work on Physics based deep learning has found success in learning a latent linear representation of the dynamics in a latent feature space. While MLPs can model the relational information between features as an undirected and fully connected graph, a recurrent or autoregressive learns a causal graph between them. In a particle based system, such a causal graph can encode numerous dependencies between state space variables. Fig. 1 illustrates the common causal relations learned by sequential models and the structural differences in graphs encoded in the latent space. The key difference between using an autoregressive and non-autoregressive graph to map a state space to a latent space condenses between mapping a particle's single $k \times d$ dimensional feature vector to a latent space instead of mapping $k$ embeddings. Each of these additional embeddings capture a different causal relation between state space variables. Our central hypothesis therefore is that latent space encoded with a causal graph enables particle based ML surrogate models to learn long range dependencies that obey conservation principles much better than an undirected graph. Therefore, in the absence of a structure on the state space variables, inductive biases such as the next states of a system of particles affecting the initial states are learned, when in practice, such scenarios are not encountered when learning to approximate the dynamics of a forward simulation problem.

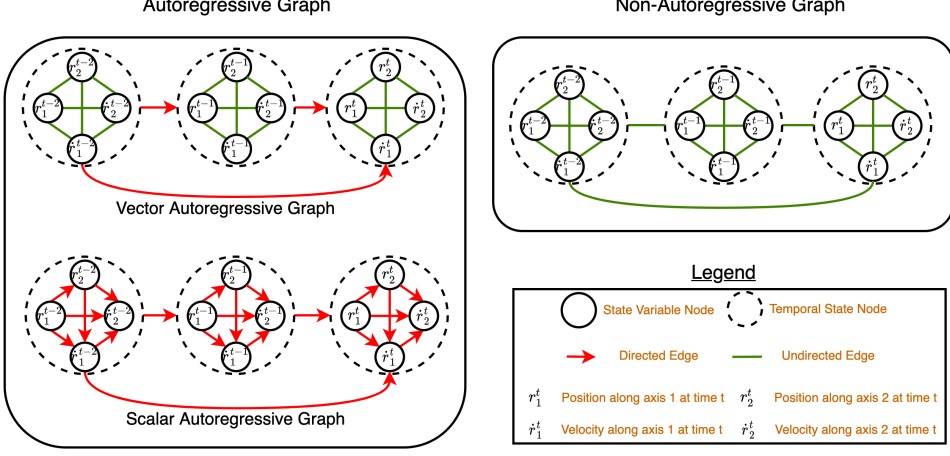

Figure 1: Structure of the state space with/without auto-regressive property. The temporal state nodes constitute an asymmetric or a DAG-like temporal graph which further contains a state space graph within each node. The directed edges do not allow interactions between the previous and the next state-variables, while the undirected edges allows such interactions.

The key contributions of the proposed approach are as follows:

- An *Autoregressive Graph Encoder* (AGN) that explicitly induces the arrow of time (i.e. previous states affect the future states) bias to capture causal relations between state space variables on the latent space.

- The induced temporal bias enables the Graph Network to achieve superior energy conservation and roll-out error accumulation performance across long time steps.

- Comparable multi-step prediction performance against a Graph Transformer model whilst requiring an order of magnitude lower parameters.

## 2 RELATED WORK

Engineering simulations developed for prediction and control of complex physical systems can be built based on empirical or theoretical findings. Compared to analytical models, a learned simulator can be more efficient at predicting complex phenomena (He et al. (2019); Sanchez-Gonzalez et al. (2020)). Further, domain knowledge can be encoded as physics priors in the form of loss function regularization (Raissi et al. (2019)) and data featurization to improve accuracy and performance. However, explicitly providing knowledge may limit generalizability to different datasets that are fundamentally governed by the same underlying dynamics. Data-driven approaches learn the dynamics of a system without explicitly solving an ODE or a PDE. The physical interactions between particles are learned implicitly by nature of the model construction (Battaglia et al. (2018); Chen et al. (2018); Greydanus et al. (2019); Cranmer et al. (2020a;b); Sanchez-Gonzalez et al. (2020); Pfaff et al. (2021); Rubanova et al. (2022)), therefore enabling generlization across a similar class of problems. Such neural networks have shown to approximate solutions of ordinary differential equations (ODE) that govern the dynamics of a system and have also shown to be compatible with most off-the-shelf ODE explicit solvers (i.e., Euler, Runge-Kutta). While successful, such meta-models have only been trained to learn the forward dynamics of the system based on the current state information or by concatenating history of states.

Deep implicit models (Amos & Kolter (2017); Bai et al. (2019; 2020)) are a powerful class of implicit learning methods proposed for learning constraints as well as learning to solve constraints. Often these models work around the computational expense of computing first and higher order gradients required by numerical optimization solvers such as gradient descent, Newton's method, .etc. Classical mechanics present a mechanistic understanding of reality that can be captured using such implicit or constrained models. Symmetries such as conservation of energy and momentum can be implicitly learned by constraining the gradients of the networks to satisfy Euler-Lagrange or Hamilton's equations (Chen et al. (2018); Lutter et al. (2019); Greydanus et al. (2019); Cranmer et al. (2020a)). Recent work (Yang et al. (2020); Rubanova et al. (2022)) have proposed implicit neural network strategies to learn physics-based constraints. These methods frame the forward problem as a constrained optimization problem. The Neural Projections method iteratively proposes a prediction of the next state of a system with an explicit Euler step to solve a root-finding problem, then projects the prediction onto a learned constraint manifold that is implemented as a multilayer perceptron (MLP). A recent work (Han et al. (2022)) proposes a sequential model with a transformer operating in the latent space to perform autoregression on latent temporal embeddings of particles. While their approach helps manage the dimensionality of the problem so as to limit memory consumption of the sequence model, it still requires $t$ sequential computations.

In this work, we induce an arrow of time constraint on the encoder parameterization by modifying it as an autoregressive encoder (Germain et al. (2015)) that encodes the nodes and edges separately with modifications to suit $n$ dimensional physics problems. The Masked Autoencoder for Distribution Estimation (MADE) model proposed by (Germain et al. (2015)) only enforces a scalar autoregressive property that may not suit vectors denoting a single state of a particle. By changing the manner in which the masking scheme is implemented, we generalize the MADE model to have a vector autoregressive property that can be used to generate temporal latent node and edge embeddings. Put together with a message passing network, we show that our model is able to model time dependencies better and lead to slower error accumulation over long time-steps.

## 3 APPROACH

### 3.1 LEARNED SIMULATOR OVERVIEW

The dynamics of a system of $N$ particles can be described by a coupled system of Ordinary Differential Equations (ODEs) that takes the following general form:

$$\frac{d}{dt}\mathbf{x}(t) = \mathbf{f}(\mathbf{x}(t), t, \boldsymbol{\omega}) \tag{1}$$

where, $\mathbf{x}(t) = [\mathbf{r} \ \dot{\mathbf{r}}]^T$ represents the state of the system at time $t$ with $\mathbf{r} \in \mathbb{R}^{N \times d}$ denoting the position vectors and $\dot{\mathbf{r}} \in \mathbb{R}^{N \times d}$ denoting the velocity vectors of $N$ particles in $d$ dimensions. The particles also possesses a physical property such as mass, charge, density, etc. and is denoted by

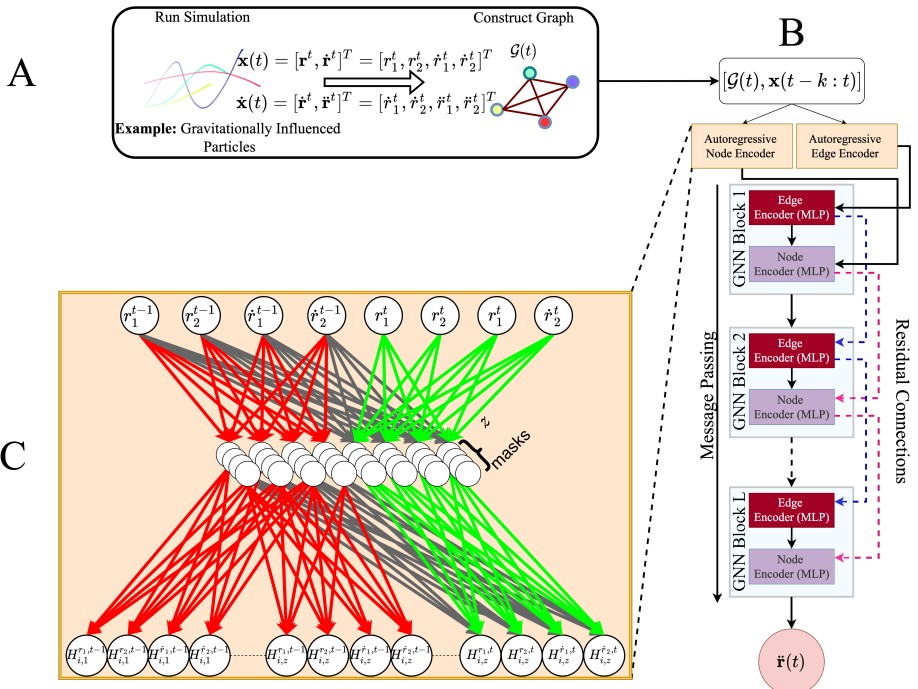

Figure 2: Schematics: **A)** Illustrates data and corresponding particle graph generation, **B)** Vector Autoregressive Graph Network, **C)** A single layer Vector Autoregressive Node Encoder with $z$ masks in the output layer that generates a $4z$ dimensional latent vector for every node $i$. The directed colored arrows denote dependence.

$\boldsymbol{\omega} \in \mathbb{R}^N$. In general this problem requires an approximate numerical solution that is acquired by solving the ODE subject to the initial value condition $\mathbf{x}(0)$. In this paper, we are interested in learning the dynamics of the system ($\mathbf{f}$) which can typically be complex using a parameterized function approximator ($\mathbf{f}_{(\cdot)}$), where ($\cdot$) denotes the parameters to be learned. Further, since we adopt a graph-based representation of the system of particles, we consider particles as nodes ($V$) in a graph ($\mathcal{G}$) such that every particle has a first-order neighborhood as well as hops of higher-order neighborhoods. Hence, every system state ($\mathbf{x}(t)$) has a corresponding adjacency matrix ($\mathbf{A}(t)$) which is a matrix representation of $\mathcal{G}(t) = (V, E)$.

To solve the forward problem, the learned simulator $\mathbf{f}_\theta : \mathbf{x}(t) \mapsto \dot{\mathbf{x}}(t)$ maps the current state of the system of particles at time $t$ to its first-order derivative which is the velocity and acceleration vectors of the system of particles. We note that the training is supervised on acceleration labels only. Then, using a choice of numerical integrator (i.e., Euler, Runge-Kuta) the next state $\mathbf{x}(t+1)$ can be obtained from $\dot{\mathbf{x}}(t) = [\dot{\mathbf{r}} \ \ddot{\mathbf{r}}]^T$.

## 3.2 Learning Forward Dynamics

### 3.2.1 Vector Autoregressive (VAR) Encoder

Following the established notations, we implement the learnable function $\mathbf{f}_\theta$ using Graph Networks (GN) (Battaglia et al. (2018)), arranged in the Encode-Process-Decode architecture, similar to previous work on GN-based learned simulators (Sanchez-Gonzalez et al. (2020); Pfaff et al. (2021)). In this paper, we generalize the notion of autoregression on scalars to vectors via Vector autoregression (VAR). Specifically, they explicitly induce the notion of arrow of time wherein the future states do not affect the past states. However, a non-auto-regressive graph captures dependencies wherein the past and future states affect each other, which is typically not the case in forward simulation problems. As a result, the network as a whole tends to learn biases that may not hold beyond training. In the simple case of a 2D problem, the motion of each particle in a trajectory can be described through a vector ($[\mathbf{r}_1, \mathbf{r}_2, \dot{\mathbf{r}}_1, \dot{\mathbf{r}}_2]^T$) in 4 dimensional state space. Each element of the vector is related

to one another as well as to the next state's vector through some unknown coefficients (eg. parameterized as Neural Nets). This can compactly be captured using a linear model such as VAR. In the example below, we show a multivariate VAR model of order 2 with a noise term ($\epsilon^t$).

$$
\begin{bmatrix} r_1^t \\ r_2^t \\ \dot{r}_1^t \\ \dot{r}_2^t \\ r_1^{t-1} \\ r_2^{t-1} \\ \dot{r}_1^{t-1} \\ \dot{r}_2^{t-1} \end{bmatrix} = \begin{bmatrix} w'_{1,1} & w'_{1,2} & w'_{1,3} & w'_{1,4} & w'_{1,5} & w'_{1,6} & w'_{1,7} & w'_{1,8} \\ w'_{2,1} & w'_{2,2} & w'_{2,3} & w'_{2,4} & w'_{2,5} & w'_{1,6} & w'_{1,7} & w'_{1,8} \\ w'_{3,1} & w'_{3,2} & w'_{3,3} & w'_{3,4} & w'_{3,5} & w'_{1,6} & w'_{1,7} & w'_{1,8} \\ w'_{4,1} & w'_{4,2} & w'_{4,3} & w'_{4,4} & w'_{4,5} & w'_{1,6} & w'_{1,7} & w'_{1,8} \\ w'_{5,1} & w'_{5,2} & w'_{5,3} & w'_{5,4} & 0 & 0 & 0 & 0 \\ w'_{6,1} & w'_{6,2} & w'_{6,3} & w'_{6,4} & 0 & 0 & 0 & 0 \\ w'_{7,1} & w'_{7,2} & w'_{7,3} & w'_{7,4} & 0 & 0 & 0 & 0 \\ w'_{8,1} & w'_{8,2} & w'_{8,3} & w'_{8,4} & 0 & 0 & 0 & 0 \end{bmatrix} \begin{bmatrix} r_1^{t-1} \\ r_2^{t-1} \\ \dot{r}_1^{t-1} \\ \dot{r}_2^{t-1} \\ r_1^{t-2} \\ r_2^{t-2} \\ \dot{r}_1^{t-2} \\ \dot{r}_2^{t-2} \end{bmatrix} + \begin{bmatrix} \epsilon_1^t \\ \epsilon_2^t \\ \epsilon_3^t \\ \epsilon_4^t \\ 0 \\ 0 \\ 0 \\ 0 \end{bmatrix} \tag{2}
$$

In Eqn. 2, $\mathbf{W'} = \mathbf{W} \circ \mathbf{M^W}$, where $\mathbf{W}$ represents a parameterizable autoregressive coefficient matrix such that elements $\{w'_{5:8,1:4}\} = 1$ and $\mathbf{M^W}$ is a mask the size of $\mathbf{W}$ that can be used to induce the desired dependencies in a generalizable manner between input and output variables of interest. In the case above, the matrix $\mathbf{M^W}$ creates the following temporal dependencies: $\mathbf{r}^t | (\mathbf{r}^{t-1}, \mathbf{r}^{t-2})$; $\mathbf{r}^{t-1} | \mathbf{r}^{t-1}$. Since we seek to parameterize such a VAR process using a neural network, we show a similar analogy in the context of a an autoencoder with a single hidden layer. We drop the noise term in our NN formulation. Let the output of the first hidden layer be denoted as $\mathbf{H}(\mathbf{x}) = act(\mathbf{b} + (\mathbf{W} \circ \mathbf{M^W})\mathbf{x})$, which is further used in computing the prediction ($\hat{\mathbf{x}}$), such that $\hat{\mathbf{x}} = act(c + (\mathbf{V} \circ \mathbf{M^V})\mathbf{H}(\mathbf{x}))$. In the expressions above, $act$ implies an activation function (typically ReLU), $\mathbf{M^W}$ and $\mathbf{M^V}$ are the masks that impose the autoregressive property while $\mathbf{V}$ and $\mathbf{W}$ are the weight matrices that parameterize the autoregressive coefficient matrix for a VAR based encoder of order $k$..

To design a set of masks that preserve the expected input-output relations, we begin by assigning each element $x$ of the $d$ dimensional state vector $\mathbf{x}(t)$ an integer $t$ corresponding to the $t^{th}$ time step and each hidden unit $s$ in the hidden layer an integer $p$ as a function ($m(s,t)$) of $s$ and $t$. Hence, the $s^{th}$ hidden unit's dependence can be computed as follows: $m(s,t) = s \mod d + (d - (s \mod d)) + dt = d(1 + t) = p$ gives the maximum number of input units to which it can be connected. Further, we allow $m(s,t) = P$, where $P = kd$, while not allowing $m(s,t) = 0$ so as to ensure that the current state as well as the $k^{th}$ lag state are conditioned upon when generating high-dimensional temporal latent vectors as opposed to probabilisitic conditionals or prior parameterization in generative models.

The conditional temporal latent vectors can be generated using a deep autoencoder parameterized with the help of the following masking rules. In the single hidden layer case, the constraints on the maximum number of inputs to each hidden unit are encoded in the matrix masking the connections between the input and hidden units:

$$
\mathbf{M}_{s,p}^{\mathbf{W}} = \begin{cases} 1_{m(s,t) \geq p}, & \text{if } m(s,t) \geq p \\ 0, & \text{otherwise} \end{cases} \tag{3}
$$

for $p \in \{d, 2d, \ldots, kd\}$, $s \in \{1, \ldots, S\}$ and $t \in \{0, \ldots, k-1\}$. Further, we encode the constraint that the $p^{th}$ network output is connected to all $m(s,t) = p$ so that the output weights can connect the $p^{th}$ network output to hidden units with $m(s,t) \leq p$. These constraints can be encoded in the output mask matrix as follows:

$$
\mathbf{M}_{p,s}^{\mathbf{V}} = \begin{cases} 1_{p \geq m(s,t)}, & \text{if } p \geq m(s,t) \\ 0, & \text{otherwise} \end{cases} \tag{4}
$$

The proposed encoder architecture naturally generalizes to deep architectures. To impose the same constraint on the succeeding hidden layer, we just have to ensure that each hidden unit $s'$ in the subsequent layer gets connected to the preceeding layers such that $m^l(s,t) \leq m^{l+1}(s',t)$, where $l$ denotes the layer number. The rule given by Eqn. (3) can be generalized to any layer $l$. The last hidden layer ($l'$) should preserve the natural ordering (vector autoregressive graph) by enforcing Eqn. (4). Further, we are interested to generate latent temporal embeddings of varying size ($z$). To do so, we create $z$ copies of final layer masks such that each mask generates an embedding the size of the input vector (i.e. if the input of node $i$ has a size of $|\mathbf{x}_i|$, then the vector autoregressive

output of the encoder will be of size $z \cdot |\mathbf{x}_i|$. We pick a $z$ value by matching the encoder output size with that of a typical autoencoder implemented in a GN.

We have a vector autoregressive node encoder ($\mathcal{VAR}_V$) and an edge encoder ($\mathcal{VAR}_E$) whose corresponding output node and edge embeddings are denoted as $\mathbf{H}_i^{(0)}$ and $\mathbf{H}_{ij}^{(0)}$ respectively. The input to $\mathcal{VAR}_E$ is $[\mathbf{x}_i(t-k) - \mathbf{x}_j(t-k), \dot{\mathbf{x}}_i(t-k) - \dot{\mathbf{x}}_j(t-k), \dots, \mathbf{x}_i(t) - \mathbf{x}_j(t), \dot{\mathbf{x}}_i(t) - \dot{\mathbf{x}}_j(t)]$ representing the relative position and velocity information between particles across $k$ (hyper-parameter) time-steps, while that of $\mathcal{VAR}_V$ is $[\mathbf{r}_i(t-k), \dot{\mathbf{r}}_i(t-k), \dots, \mathbf{r}_i(t), \dot{\mathbf{r}}_i(t)]$ representing the full state vector between particles across $k$ (hyper-parameter) time-steps.

### 3.2.2 Processor and Decoder

Having described the graph encoders, we now briefly describe the graph processor and decoder along with their respective inputs and outputs. The graph processor computes interactions among nodes through $L$ message passing blocks arranged in sequence using standard MLP edge encoders $\mathcal{E}_E$ and node encoders $\mathcal{E}_V$. Message-passing preserves constraints while allowing information to propagate between nodes and edges. The processor takes as input the latent temporal embeddings ($\mathbf{H}_i^{(0)}$ and $\mathbf{H}_{ij}^{(0)}$) generated by the autoregressive node and edge encoders to perform message passing computations. In the subsequent sections, we drop $t$ for brevity.

The edge encoder ($\mathcal{E}_E$) performs the following node-to-edge ($V \to E$) message passing operation,

$$V \to E : \mathbf{H}_{ij}^{(l)} = \mathcal{E}_E^{(l)}([\mathbf{H}_i^{(l)} - \mathbf{H}_j^{(l)}, |\mathbf{H}_i^{(l)} - \mathbf{H}_j^{(l)}|, \mathbf{H}_{ij}^{(l-1)}]) + \mathbf{H}_{ij}^{(l-1)} \tag{5}$$

where, $[.\,,.]$ denotes concatenation and $l$ denotes the $l$-th graph network block. Following the edge encoding operation, the node encoder ($\mathcal{E}_V$) takes as input the edge embedding and performs the following edge-to-node ($E \to V$) message passing operation,

$$E \to V : \mathbf{H}_i^{(l+1)} = \mathcal{E}_V^{(l)}([\sum_{j \in \mathcal{N}_i} \mathbf{H}_{ij}^{(l)}, \mathbf{H}_i^{(l)}]) + \mathbf{H}_i^{(l)} \tag{6}$$

The node encoder aggregates the edge features for a target node $i$ across its neighboring nodes ($\mathcal{N}_i$), concatenates the target node features and finally adds the previous block's latent graph features using residual connections. Following the processing step, the output of the last message passing block gets passed to an MLP decoder that predicts the acceleration ($\hat{\ddot{\mathbf{r}}}$). See Supplementary Materials for more details on implementation.

## 4 Discussion

In this section, we discuss the key findings from our experiments. In order to evaluate the rigorousness of the proposed approach, we keep a number of hyper-parameter choices such as the number of hidden layers, hidden neurons, etc. constant. See Supplementary Materials for full details on datasets and choice of hyper-parameters. We benchmark our approach against the following work on one-step forward prediction: Graph Network (GN) (Sanchez-Gonzalez et al. (2020)), Graph Lagrangian Network (GLN) (Cranmer et al. (2020a)), Graph Hamiltonian Network (GHN) (Greydanus et al. (2019); Cranmer et al. (2020b)), Graph Transformer Network (GTN) (Shi et al. (2020). We also benchmark against sequential forward prediction models such as Gated Graph Recurrent Neural Networks (GGRNN) (Seo et al. (2018)) and by adapting the base Graph Transformer architecture of Haan et al. Han et al. (2022) without regularizing the latent space nor pre-computing the latent embeddings. While Pfaff et al. (2020) propose a GN for learning physics simulations, we note that their approach when applied to particle based datasets simply assumes the architecture as proposed by Sanchez-Gonzalez et al. (2020). See Supplementary Materials for more details on time-complexity analysis, model parameter count and baseline implementations.

### 4.1 Performance of forward dynamics prediction

We train all baselines (except the sequential models) as well as our models on one-step target acceleration predictions (i.e., Given $\mathbf{x}(t : t + k)$, predict $\ddot{\mathbf{r}}(t + k)$) using a 70:30 train, test ratio. Once trained, we sample 30 initial conditions that were not a part of the train/test distribution to generate 30 simulations over a 1000 time-step roll-out using an RK4 integrator.

The optimum lookback length ($k^*$) is a function of the model as well as the dataset. While we discuss the impact of $k$ on the stability and forward prediction performance later, tables 1 and 4 report the Mean Squared Error (MSE) on multi step predictions when optimum $k^*$ is chosen for each model/dataset. See Supplementary Materials for more details on look back length analysis. We impose translation invariance by construction for all baseline models by explicitly not providing absolute positions as input. Through our experiments, we find that our approach performs substantially better when provided with absolute positions while baselines perform worse with absolute positions as input. While this is surprising, we provide a reasoning for why this is the case. Please refer to the partial time-translation analysis in the supplementary material for more details.

### 4.1.1 FORWARD SIMULATION COMPARISON WITH SEQUENTIAL MODELS

We arbitrarily fix $k = 5$ and compare AGN, along with other one-step and multi-step methods. We use GTN to perform both single-step predictions (no temporal recurrence) as well as multi-step predictions under the many-to-many setting wherein we input $k$ states and predict $k$ acceleration targets corresponding to each state. Table 2 shows the forward simulation performance across the methods with GTN (MS) outperforming the AGN (scalar) and AGN (vector) on 6 out of 10 datasets. Despite the slightly better performance of GTN (MS), AGN has an order of magnitude fewer parameters to learn and therefore can be trained faster to achieve GTN (MS) like performance. When comparing GTN (MS) with GTN (SS), the performance of GTN (MS) is substantial across all datasets except on 2D/3D Gravity ($\frac{1}{r^2}$). This difference in performance can clearly be attributed to the temporal biases learned by a temporal model that are simply not captured by a non-temporal model. However, in comparison to GTN (SS), AGN performs substanially better across the board. While GGRNN being a temporal model, it's performance is quite poor across the datasets and is outperformed by all single step methods that have been considered in this paper. This difference in performance can likely be attributed to the better expressiveness of the other methods (temporal and non-temporal). While we hypothesize that AGN could possible show comparable performance with GTN (MS) on different $k$, we instead analyze the energy conservation property of the models in the next section. See Supplementary Materials for additional forward prediction comparisons.

Table 1: Mean roll-our error comparison between one-step methods for an optimum $k^*$

| Datasets | GN | GTN | AGN (vector) (Ours) | AGN (scalar) (Ours) |
|---|---|---|---|---|
| 2D Spring | 5.38 | 3.49 | 1.77 | **1.36** |
| 2D Damped | 9.37 | 11.04 | 8.17 | **7.25** |
| 2D Gravity ($\frac{1}{r}$) | 10.55 | 165.39 | 9.41 | **6.95** |
| 2D Gravity ($\frac{1}{r^2}$) | 5.12 | 5.62 | 3.58 | **2.84** |
| 2D Charge | 3.85 | **1.26** | **1.26** | 1.77 |
| 3D Spring | 10.57 | **4.76** | 7.54 | 8.55 |
| 3D Damped | 16.13 | 21.52 | 13.46 | **12.23** |
| 3D Gravity ($\frac{1}{r}$) | 2.19 | **0.45** | 7.38 | 9.97 |
| 3D Gravity ($\frac{1}{r^2}$) | 0.88 | 0.91 | 0.79 | **0.70** |
| 3D Charge | 0.94 | 0.47 | 0.47 | **0.46** |

Table 2: Roll-out error comparison between sequential and non-sequential models when $k = 5$. SS-Single Step, MS-Multi Step

| Datasets | GN (SS) | GTN (SS) | GTN (MS) | GGRNN (MS) | AGN (vector) (SS) (Ours) | AGN (scalar) (SS) (Ours) |
|---|---|---|---|---|---|---|
| 2D Spring | 6.72 | 3.49 | 3.88 | 235.02 | 2.68 | **1.36** |
| 2D Damped | 19.39 | 977.89 | **6.89** | 709.17 | 12.76 | 7.25 |
| 2D Gravity ($\frac{1}{r}$) | 14.96 | 165.39 | **4.67** | 33.78 | 27.58 | 17.59 |
| 2D Gravity ($\frac{1}{r^2}$) | 299.76 | 569.31 | 4781.57 | 61.30 | 3.58 | **3.24** |
| 2D Charge | 38.06 | 15.78 | 2.17 | 5.15 | 8.08 | **1.77** |
| 3D Spring | 18.09 | 6.34 | **2.04** | 184.02 | 10.75 | 16.17 |
| 3D Damped | 202.71 | 251.98 | **13.92** | 887.01 | 15.73 | 20.26 |
| 3D Gravity ($\frac{1}{r}$) | 2.69 | 1.26 | **0.55** | 224.04 | 7.38 | 19.58 |
| 3D Gravity ($\frac{1}{r^2}$) | 1.42 | 2.91 | 808.004 | 1.70 | **1.16** | 1.60 |
| 3D Charge | 3.82 | 7.52 | **0.47** | 6.47 | 1.05 | **0.47** |

### 4.1.2 ENERGY CONSERVATION COMPARISON WITH SEQUENTIAL MODELS

Roll-out errors unfortunately do not narrate the entire story. Since energy and roll-out error share an intimate relationship, we compute and report (see table 3) the energy MSE between the predicted trajectories and the ground-truth trajectories by measuring the total energy deviation. While we notice that GTN's (MS) roll-out error is superior across the datasets, we do not notice a similar trend when comparing their corresponding energy MSEs. We find AGN models, specifically the scalar AGN (SS) model to outperform GTN (MS) by conserving energy more efficiently across the trajectory. This difference is due to the fact that GTN (MS) utilizes a linear encoder that does not capture conditional relationships between states (i.e., $\mathbf{x}^t$ is not dependent on any of $\mathbf{x}^{<t}$). However, as a result of AGN models capturing such relations, we notice that it conserves energy much better than all baselines, especially on non-conservative systems (2D/3D damped spring). We hypothesize that as energy continues to accrue, the roll-out MSE of GTN (MS) may reach higher-values than AGN if the simulation is continued for longer time-steps. Figs.3 and 4 further illustrate the difference in roll-out and energy MSE for different $k$ on a non-conservative and conservative system

respectively. While we expected to see better prediction performance by the AGN (vector) model due to their added structure that captures state space relations, the simple and slightly less structured AGN (scalar) consistently outperforms the AGN (vector) model. See Supplementary Materials for additional energy analysis on single-step methods.

Table 3: Roll-out energy accumulation error comparison between sequential and non-sequential models when $k = 5$. SS-Single Step Model, MS-Multi Step Model; - indicates an extremely large number ($>$ than 1e7)

| Datasets | GN (SS) | GTN (SS) | GTN (MS) | GGRNN (MS) | AGN (vector) (SS) (Ours) | AGN (scalar) (SS) (Ours) |
|---|---|---|---|---|---|---|
| 2D Spring | 790.86 | 354.46 | 1662.14 | - | 399.19 | **206.04** |
| 2D Damped | 208.38 | 749412.46 | 545.97 | - | **55.93** | 371.3 |
| 2D Gravity ($\frac{1}{r1}$) | 1884.96 | 83541.03 | **153.58** | 1181.45 | 2406.93 | 903.16 |
| 2D Gravity ($\frac{1}{r2}$) | - | - | - | 939246.92 | 46334.59 | **3181.42** |
| 2D Charge | 65947.64 | 266108.12 | **44.21** | 9053.59 | 190091.34 | 236.87 |
| 3D Spring | 701.13 | **255.47** | 734.28 | - | 1294.72 | 563.2 |
| 3D Damped | 3714.71 | 21788.21 | 226.74 | - | 401.413 | **4.51** |
| 3D Gravity ($\frac{1}{r1}$) | 604.51 | 365.97 | **325.60** | 792144.001 | 2345.27 | 3730.01 |
| 3D Gravity ($\frac{1}{r2}$) | 4771.97 | 72661.08 | - | **483.10** | 15742.91 | 106459.40 |
| 3D Charge | 21107.90 | 125326.42 | 2.92 | 22441.39 | 272.09 | **2.15** |

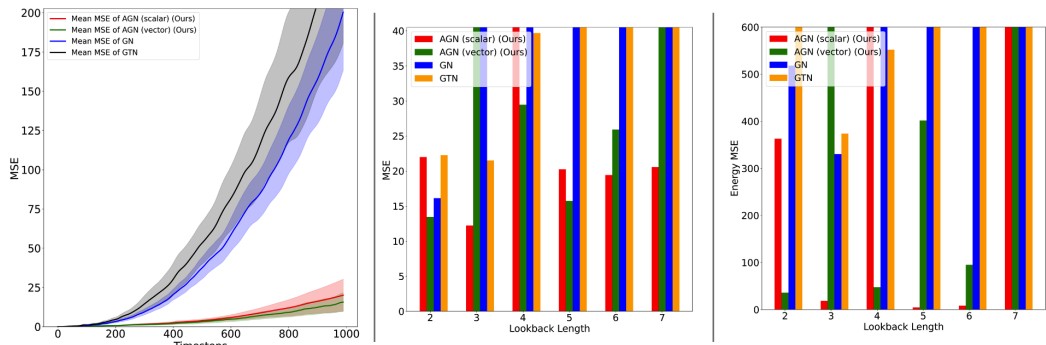

Figure 3: (3D Damped) is a non-conservative system. Left: Roll-out error across models when $k = 5$. Middle: Roll-out error look back comparison. Right: Roll-out energy accumulation look back comparison. We crop the Y axis of the bar graphs using the median MSE across models for better visualization.

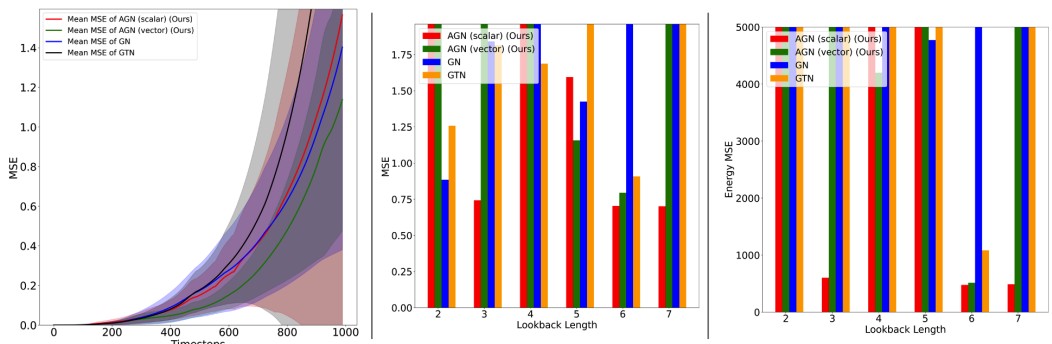

Figure 4: (3D Gravity ($r2$)) is a conservative system. Left: Roll-out error across models when $k = 5$. Middle: Roll-out error look back comparison. Right: Roll-out energy accumulation look back comparison. We crop the Y axis of the bar graphs using the median MSE across models for better visualization.

### 4.1.3 PHYSICS CONSTRAINTS

Next, we perform experiments by modifying a GN and AGN to have physics biases (Lagrangian and Hamiltonian). Since these constraints require computing the gradients with respect to $r^t$ and

momentum at time $t$, they require absolute positions of the particles as input. Like AGN, we employ an autoregressive node and edge encoder to capture spatio-temporal latent embeddings that are then used as input to approximate the Lagrangian and Hamiltonian of the parameteric learnable function ($f_{(.)}$). From table 4, we can see that AGLN (vector and scalar variants) outperforms the baseline Graph Lagrangian Network in 8 out of 10 datasets while AGHN (vector and scalar variants) outperforms the baseline Graph Hamiltonian Network in 9 out of 10 datasets when trained with the optimum $k$. Contrary to the dominating performance of AGN (scalar) variant in the earlier comparison, AGLN/AGHN (vector) model seems to outperform AGLN/AGHN (scalar) variant in a majority of the cases. This shows that physics-induced models are able to exploit the state space structure to gain advantage over non-physics models. Unlike the non-physics induced models, the major argument underlying the success of Lagrangian and Hamiltonian networks is that they are able to conserve energy much more efficiently and for longer time-steps. To validate how well the proposed physics-induced models conserve energy, we perform a similar energy analysis (see Table 5) on the trajectories predicted by all physics-induced models. Specifically, we report the energy MSE corresponding to the optimum $k$, therefore allowing us to infer its relationship with the roll-out MSE. In general, we find that GLN and AGLN conserve energy much better than GHN, AGHN and non-physics induced models. While there seems to be a tie between AGLN and GLN models in terms of energy MSE, we note that GLN is subject to a higher deviation in energy accumulation, as observed from its performance on the 2D/3D charge and 3D Gravity ($\frac{1}{r^2}$) datasets. Hence, we find that autoregressive models do indeed help already physics-biased models to preserve energy over long time steps. We report additional results for choice of other $k$ values in the supplementary material. During training, we notice that both GLN and GHN are sensitive to choice of lookback length and a sub-optimal lookback length affects the prediction performance as well as causes a blow-up of error (instability). On the contrary, we find that our AGLN and AGHN models like their AGN counterpart are quite stable even when using a sub-optimal lookback length.

Table 4: Mean roll-out error comparison across Physics Constrained GNs for an optimum $k^*$

| Datasets | GLN | GHN | AGLN (vector) (Ours) | AGLN (scalar) (Ours) | AGHN (vector) (Ours) | AGHN (scalar) (Ours) |
|---|---|---|---|---|---|---|
| 2D Spring | 0.84 | 2.05 | **0.70** | 0.89 | 2.23 | 1.27 |
| 2D Damped | 6.64 | 426.62 | 3.29 | **2.31** | 16.09 | 8.88 |
| 2D Gravity ($\frac{1}{r}$) | **2.94** | 13.64 | 18.16 | 7.62 | 11.08 | 26.37 |
| 2D Gravity ($\frac{1}{r^2}$) | 5.70 | 52.78 | **1.01** | 1.99 | 1.29 | 6.84 |
| 2D Charge | 20.68 | 42.74 | **0.98** | 3.61 | 4.04 | 5.86 |
| 3D Spring | **0.64** | 1.76 | 1.37 | 0.95 | 10.08 | 4.49 |
| 3D Damped | 5.43 | 46.62 | 20.06 | **4.58** | 85.62 | 24.97 |
| 3D Gravity ($\frac{1}{r}$) | 2.43 | 5.52 | **1.62** | 1.68 | 3.86 | 6.90 |
| 3D Gravity ($\frac{1}{r^2}$) | 9.24 | 1.41 | **0.72** | 0.86 | 0.75 | 1.40 |
| 3D Charge | 37.11 | 1.92 | 0.95 | 0.82 | **0.44** | 0.85 |

Table 5: Mean energy accumulation comparison across Physics Constrained GNs for an optimum $k^*$

| Datasets | GLN | GHN | AGLN (vector) (Ours) | AGLN (scalar) (Ours) | AGHN (vector) (Ours) | AGHN (scalar) (Ours) |
|---|---|---|---|---|---|---|
| 2D Spring | **140.04** | 193.04 | 274.46 | 232.19 | 724.51 | 801.11 |
| 2D Damped | 29.04 | - | 44.73 | **7.39** | 333.88 | 80.68 |
| 2D Gravity ($\frac{1}{r}$) | 171.84 | 485.29 | **127.83** | 177.53 | 1160.62 | 1141.59 |
| 2D Gravity ($\frac{1}{r^2}$) | 163.15 | 119309.05 | **139.90** | 164.13 | 211.10 | 837.81 |
| 2D Charge | 10443.70 | 67054.94 | **8.51** | 192.64 | 350.67 | 303.63 |
| 3D Spring | **246.62** | 413.83 | 546.60 | 614.45 | 132415.14 | 26054.92 |
| 3D Damped | **109.23** | 25054.80 | 158.51 | 207.91 | 102828.64 | 167665.90 |
| 3D Gravity ($\frac{1}{r}$) | **88.40** | 1297.12 | 262.81 | 150.54 | 318.83 | 3742.82 |
| 3D Gravity ($\frac{1}{r^2}$) | 4080.84 | 572.53 | **453.04** | 471.41 | 474.49 | 505.94 |
| 3D Charge | 24641.89 | 151.86 | 3.26 | 11.55 | **2.07** | 7.28 |

## 5 CONCLUSION

In this paper, we introduce a Graph Network with a temporal inductive bias to learn forward physics simulations. The temporal bias is enforced by an Autoregressive encoder that attends to the previous state inputs of a system of particles. The proposed model circumvents sequential computations required to compute latent vectors and therefore does not sacrifice speed for accuracy and instead allows a GN to enjoy the best of both worlds. Further, by capturing diverse state relations from a trajectory of length $k$, it offers the best roll-out and energy MSE performances. This bias enables our model to perform robust multi-step predictions when compared against the baselines.

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
