# OpenReview forum: "Autoregressive Graph Network for Learning Multi-step Physics"
_ICLR.cc/2023/Conference — Submitted to ICLR 2023_

### Official Review · Reviewer_ucoP · 2022-10-21

**Confidence:** 3
**Correctness:** 3
**Technical Novelty And Significance:** 1
**Empirical Novelty And Significance:** 2
**Recommendation:** 3

**Clarity, Quality, Novelty And Reproducibility:**

The paper is clearly written.
As discussed in "Weaknesses", I don't think the proposed method is particularly novel.
I am also not sure if it could be reproduced from the details proposed in the supplementary material. I also wonder if more details could be provided about the Lagrange and Hamiltonion GN baselines that the authors compared their method to.

**Strength And Weaknesses:**

Strengths:
1. The proposed method is extensively evaluated, with many experiments on a range of tasks.

Weaknesses:
1. The proposed method does not feel especially novel. One of the main contributions discussed, the concept of "vector autoregression" feels like a trivial extension of existing work, either of [1] by extending it to be autoregressive over time, or of [2] by extending it to particle systems via a GN.
2. Some of the arguments about benefits of the proposed method feel tenuous at best. For example, I have a hard time following the discussion about translation invariance. The authors claim that "By providing absolute positions as input, the autoregressive encoder learns translation invariant features." -> Why exactly does this happen?


[1] Learning to Simulate Complex Physics with Graph Networks. Sanchez-Gonzalez et al, https://arxiv.org/abs/2002.09405.
[2] Prediction and control with temporal segment models. Mishra et al, https://arxiv.org/abs/1703.04070.

**Summary Of The Paper:**

The authors propose a new graph network (GN) architecture for learning dynamics models of particle-based systems. Their architecture is structured as a GN applied autoregressively in the temporal direction. They compare their method to various alternative GN architectures (vanilla, physics-inspired) on a few different dynamical systems.

**Summary Of The Review:**

Combining the concerns about novelty (see "Strengths & Weaknesses") and the sparse explanation of the baselines (see "Clarity, Quality, Novelty And Reproducibility"), I have a hard time understanding why the proposed method outperforms the baselines. Based on the provided results, it does seem to perform well empirically, but I'm wondering if there is some other feature of the proposed method (e.g. some architectural choice) or some overlooked flaw of the baselines or that explains this? Without a clear understanding of the contributions nor the empirical success, I would have a hard time recommending this paper for acceptance.

---

> ### Author Response · Authors · 2022-11-17
> **Response to Reviewer ucoP**
>
> We sincerely thank you for taking time to review our paper and provide constructive feedback. We address your comments and questions below:
>
> >The proposed method does not feel especially novel. One of the main contributions discussed, the concept of "vector autoregression" feels like a trivial extension of existing work, either of [1] by extending it to be autoregressive over time, or of [2] by extending it to particle systems via a GN.
>
> $\textbf{Response:}$ Thanks for sharing these references. The authors of [1], in their paper propose several potential directions to advance the work, notably, they discuss about the possible advantages of a learned simulator with a memory mechanism. While RNNs are theoretically more expressive and powerful than autoregressive~(AR) models, AR models trade-off model complexity for stable and parallellizable computations to name a few. On the other hand, Transformers offer the same benefits that an AR model has to offer while being as expressive as RNNs due to their model complexity. However, the only caveat due to its model complexity is that the model's time and space complexity grows quadratically.  Hence, while a linear encoder by construction may not have an internal memory mechanism, an AR encoder is a linear model that comes with a limited memory mechanism. This means, graph based physics solvers can perform inference as fast as a vanilla GN such as [1], while being able to learn dependencies across time. We have added a section on time complexity analysis and parameter count to further illustrate the differences between our model and the baselines.
>
> AR models can operate on latent space embeddings as well as directly on the state space. The key contribution of our work is that we encode a directed graph (arrow of time inductive bias) representation of the state space variables therefore capturing k latent embeddings, each capturing a conditional dependence with respect to the previous state(s). We discuss this in detail in the introduction of the updated rebuttal revision. In addition to comparison against one-step methods, we include additional experiments comparing our approach with a Gated Graph Recurrent Network (GGRNN) [2] and a Graph Transformer Network (GTN) [3,4]. We empirically find that by replacing a linear encoder with an autoregressive encoder, we obtain comparable roll-out MSE performance with GTN. Further, we perform an energy analysis to compare roll-out energy accumulation across models. We find that our model is able to preserve the principle of conservation of energy across time, outperforming multi step GTN and GGRNN. In addition, we observe a similar phenomenon across physics-induced models, wherein the Lagrangian and Hamiltonian Networks conserve energy much more efficiently with the addition of the autoregressive encoder. We hope that these points as well as our updates address your concerns regarding the novelty of our work.
>
> >Some of the arguments about benefits of the proposed method feel tenuous at best. For example, I have a hard time following the discussion about translation invariance. The authors claim that "By providing absolute positions as input, the autoregressive encoder learns translation invariant features." -> Why exactly does this happen?
>
> $\textbf{Response:}$ Thanks for raising this question. While this is not our main contribution, we were quite surprised to see that including absolute positions as input to an AR node encoder enabled it to perform roll-out predictions with a much lower MSE than by not providing them. In order to test whether we observe the same phenomenon on a vanilla GN, we conduct an experiment wherein we provide the GN with and without absolute positions. Instead we observed that in GN, performance improved when absolute positions were not given as input.  Since we do not have a theoretical analysis, based on literature, we hypothesize that AGN is partially time translation invariant since it models sequential/time-series data. We apologize for simply stating it as translation invariance. Our hypothesis is grounded on the basis of the following reasoning. Models such as Pixel Recurrent Network [5], use an autoregressive strategy to learn a generative model wherein, translation invariance is induced as a result of weight sharing. Analogous to that, we can observe that in AGN, a time shifted signal (data augmentation bias [6]) is being causally convolved with a set of fixed neural network weights, therefore inducing a partial time translation invariance. Further, we assume that in addition to relative features, positional features enable the network to learn better relational state features (where is the previous position?). Such a phenomenon can be observed while training other memory models such as RNN and Transformers, which require positional encodings. We have moved this discussion as well as the experiments to the supplementary section and toned down our claims.

---

> > ### Author Response · Authors · 2022-11-19
> > **Response to Reviewer ucoP**
> >
> > Dear Reviewer,
> >
> > Thanks again for your review. The discussion period is almost ending. We hope we have addressed your questions and concerns. Please let us know if there are anymore questions/concerns that we can address to make our paper better.

---

> ### Author Response · Authors · 2022-11-17
> **Response to Reviewer ucoP**
>
> >The paper is clearly written. As discussed in "Weaknesses", I don't think the proposed method is particularly novel. I am also not sure if it could be reproduced from the details proposed in the supplementary material. I also wonder if more details could be provided about the Lagrange and Hamiltonion GN baselines that the authors compared their method to.
>
> $\textbf{Response:}$ We hope that our additional motivation, analysis and experiments during the rebuttal revision addresses your concern regarding the novelty of our approach. We have further provided additional details on our approach as well as all the baselines, including Lagrangian and Hamiltonian GNs. In addition, we have also provided our code with this submission.
>
> $\textbf{References:}$
> [1] Sanchez-Gonzalez, A., Godwin, J., Pfaff, T., Ying, R., Leskovec, J., & Battaglia, P. (2020, November). Learning to simulate complex physics with graph networks. In International Conference on Machine Learning (pp. 8459-8468). PMLR. \
> [2] Youngjoo Seo, Michael Defferrard, Pierre Vandergheynst, and Xavier Bresson. Structured sequence modeling with graph convolutional recurrent networks. In International conference on neural information processing, pp. 362–373. Springer, 2018. \
> [3] Xu Han, Han Gao, Tobias Pffaf, Jian-Xun Wang, and Li-Ping Liu. Predicting Physics in Mesh- reduced Space with Temporal Attention. arXiv preprint arXiv:2201.09113, 2022. \
> [4] Yunsheng Shi, Zhengjie Huang, Shikun Feng, Hui Zhong, Wenjin Wang, and Yu Sun. Masked label prediction: Unified message passing model for semi-supervised classification. arXiv preprint arXiv:2009.03509, 2020. \
> [5] Aaron Van Den Oord, Nal Kalchbrenner, and Koray Kavukcuoglu. Pixel recurrent neural networks.
> In International conference on machine learning, pp. 1747–1756. PMLR, 2016. \
> [6] Valerio Biscione and Jeffrey Bowers. Learning translation invariance in CNNs. arXiv preprint
> arXiv:2011.11757, 2020.

---

### Official Review · Reviewer_xSTt · 2022-10-24

**Confidence:** 4
**Correctness:** 2
**Technical Novelty And Significance:** 2
**Empirical Novelty And Significance:** Not applicable
**Recommendation:** 3

**Clarity, Quality, Novelty And Reproducibility:**

As mentioned in the previous section, several aspects regarding motivation and architecture remain unclear. I found the explanations hard to follow, in general.

The appendix contains many graphs, but doesn't give much new information or analysis. Some figures, like figure 3, also seem to show that there's little advantage gained from the proposed method.


**Strength And Weaknesses:**

The main goal of the paper - more accurate predictions for learned physical systems - is an important and relevant one. Also, the evaluations seem to show improvements for the proposed architectures. However, I see quite a few problems and unclear points with the algorithm and manuscript:

Among others, I was surprised about the focus and attention that is given to the "autoregressive" label. This is a fairly standard approach for time integration methods, and the necessity for multiple previous steps is an inherent drawback of learned algorithms: from classical physics we know that the evolution of a deterministic physics system is uniquely prescribed by a single state. The necessity for working with multiple time steps ("lookback" as its called here) is not necessarily a positive aspect. The use of linear encoder/decoder blocks for graph networks is also not exactly new, e.g., see Stachenfeld et al. 2021 "Learned coarse models".

In addition, I was missing an explanation for why these additional linear layers should have a significant impact on the learning task (beyond adding more weights). This is likewise something that could be handled by the MLPs in the message passing layers. Here the paper also does not make clear how the weight count changes when including the proposed encoders.


**Summary Of The Paper:**

This paper proposes to add a linear node and edge encoding layer before the message passing steps of a graph network in order to improve inference performance. This encoder is employed for physics problems where the networks have the task to infer accelerations for time integration of the system under consideration, and its main job seems to be a selection of features from current and previous time step information. This approach is demonstrated for four different, but relatively simple types of particle-based systems.


**Summary Of The Review:**

Based on my current understanding of the architecture, I'm not clear whether the improved error measurements stem from more weights or inherent advantages of the encoding layers. So, based on my current understanding, I don't think I can recommend accepting this paper. I would recommend that the authors spend time to more clearly analyze and explain why their method leads to improvements, rather than long lists of graphs. This would also potentially make it possible to assess how well proposed changes would carry over to other settings, e.g., other types of physics simulations.

---

> ### Author Response · Authors · 2022-11-17
> **Response to Reviewer xSTt**
>
> We sincerely thank you for your time and for providing constructive feedback on the proposed approach. We address your comments and questions below:
>
> >Among others, I was surprised about the focus and attention that is given to the "autoregressive" label. This is a fairly standard approach for time integration methods, and the necessity for multiple previous steps is an inherent drawback of learned algorithms: from classical physics we know that the evolution of a deterministic physics system is uniquely prescribed by a single state. The necessity for working with multiple time steps ("lookback" as its called here) is not necessarily a positive aspect. The use of linear encoder/decoder blocks for graph networks is also not exactly new, e.g., see Stachenfeld et al. 2021 "Learned coarse models".
>
> $\textbf{Response:}$ We agree that autoregressive models are prevalent across Machine learning as well as other disciplines that study sequential/temporal data. We further concur with your argument on conditioning on previous states being an inherent drawback of learned algorithms. Currently, single-step Graph based physics models have empirically shown to benefit from conditioning on historical state information, particularly as it relates to learning to approximate the dynamics of a physics system. However, the main limitation with such approaches is that historical states along with the current state are considered as a single sample and are mapped to a latent space, commonly using an encoder. Much of the prior work [1,2,3] has found benefits in changing from state space coordinates where the dynamics can be coupled to a linear latent space in which the dynamics could be uncoupled by way of regularization. Such a process is closely related to Dynamic Mode Decomposition, wherein we are interested to approximate the dynamics of a system using a linearized system.
>
> While most work have used a linear encoder to achieve such a coordinate transformation, temporal state space contains important dependencies that may not be fully captured by a linear encoder, or the linear encoder may capture biases that may not generalize to new samples. In essence, a linear encoder captures the state space interactions as an undirected graph in which a plethora of dependencies exist. However, in our work, instead of learning an undirected graph, we bias the model to capture temporal graphs that capture the dependence between states as well as state space variables along the direction of time. Additionally, our encoder learns k latent embeddings, each capturing an increasing number of conditional dependencies as opposed to a single latent embedding of similar size. We have added substantial information regarding our encoder in the introduction of our revised manuscript.

---

> > ### Author Response · Authors · 2022-11-19
> > **Response to Reviewer xSTt**
> >
> > Dear Reviewer,
> >
> > Thanks again for your review. The discussion period is almost ending. We hope we have addressed your questions and concerns. Please let us know if there are anymore questions/concerns that we can address to make our paper better.

---

> > ### Comment · Reviewer_xSTt · 2022-11-24
> > **Acknowledgement of the responses**
> >
> > I want to thank the authors for the replies and explanations. Given the current assessments by myself and the other reviewers, I want to keep mine, and I can recommend that the authors address the raised points in a future revision of their work.

---

> ### Author Response · Authors · 2022-11-17
> **References**
>
> [1] Brunton, S. L., Proctor, J. L., & Kutz, J. N. (2016). Discovering governing equations from data by sparse identification of nonlinear dynamical systems. Proceedings of the national academy of sciences, 113(15), 3932-3937. \
> [2] Kutz, J. N., Brunton, S. L., Brunton, B. W., & Proctor, J. L. (2016). Dynamic mode decomposition: data-driven modeling of complex systems. Society for Industrial and Applied Mathematics. \
> [3] Lusch, B., Kutz, J. N., & Brunton, S. L. (2018). Deep learning for universal linear embeddings of nonlinear dynamics. Nature communications, 9(1), 1-10. \
> [4] Sanchez-Gonzalez, A., Godwin, J., Pfaff, T., Ying, R., Leskovec, J., & Battaglia, P. (2020, November). Learning to simulate complex physics with graph networks. In International Conference on Machine Learning (pp. 8459-8468). PMLR. \
> [5] Pfaff, T., Fortunato, M., Sanchez-Gonzalez, A., & Battaglia, P. W. (2020). Learning mesh-based simulation with graph networks. arXiv preprint arXiv:2010.03409. \
> [6] Rubanova, Y., Sanchez-Gonzalez, A., Pfaff, T., & Battaglia, P. (2021). Constraint-based graph network simulator. arXiv preprint arXiv:2112.09161. \
> [7] Stachenfeld, K., Fielding, D. B., Kochkov, D., Cranmer, M., Pfaff, T., Godwin, J., ... & Sanchez-Gonzalez, A. (2021). Learned Coarse Models for Efficient Turbulence Simulation. arXiv preprint arXiv:2112.15275. \
> [8] Youngjoo Seo, Michael Defferrard, Pierre Vandergheynst, and Xavier Bresson. Structured sequence modeling with graph convolutional recurrent networks. In International conference on neural information processing, pp. 362–373. Springer, 2018. \
> [9] Xu Han, Han Gao, Tobias Pffaf, Jian-Xun Wang, and Li-Ping Liu. Predicting Physics in Mesh- reduced Space with Temporal Attention. arXiv preprint arXiv:2201.09113, 2022. \
> [10] Yunsheng Shi, Zhengjie Huang, Shikun Feng, Hui Zhong, Wenjin Wang, and Yu Sun. Masked label prediction: Unified message passing model for semi-supervised classification. arXiv preprint arXiv:2009.03509, 2020.

---

### Official Review · Reviewer_n2LJ · 2022-10-24

**Confidence:** 3
**Correctness:** 3
**Technical Novelty And Significance:** 2
**Empirical Novelty And Significance:** 2
**Recommendation:** 5

**Clarity, Quality, Novelty And Reproducibility:**

The idea of the autoregressive model is not very novel. This paper lacks some necessary performance comparisons with other sequential models. The codes and data are provided, which is highly appreciated.

**Strength And Weaknesses:**

### Strength
1. The problem of designing graph networks for spatial-temporal simulation is important.
2. It somehow solves the problem of high computational costs caused by long-term memory.
3. It outperforms vanilla GNS while maintaining high efficiency.

### Weakness
1. The authors claim that this autoregressive model is better than other sequential models, such as RNN and Transformer, but there is no such comparison. I think the comparisons of both accuracy and efficiency are necessary. I am not fully convinced that this model is better than other commonly-used sequential models.
2. The idea of the autoregressive model seems not very novel.
3. The writing can be improved.

**Summary Of The Paper:**

This paper proposes an additional autoregressive module to the vanilla graph network-based simulator (GNS), which can outperform GNS while maintaining high efficiency. Specifically, it learns spatiotemporal embedding for simulating spatial-temporal trajectories, which somehow solves the problem of high computational costs caused by long-term memory. It conducts experiments on multiple scenarios such as gravity, spring, charge, etc.,  to demonstrate the effectiveness of the proposed method.

**Summary Of The Review:**

I think the problem is important and the proposed model seems effective, however, my main concern is about the advantage of this model over other sequential models, which is not supported by experiments.

---

> ### Author Response · Authors · 2022-11-17
> **Response to Reviewer n2LJ**
>
> We sincerely thank you for your time and for summarizing the merits of the proposed approach. We address your comments and questions below:
>
> >The authors claim that this autoregressive model is better than other sequential models, such as RNN and Transformer, but there is no such comparison. I think the comparisons of both accuracy and efficiency are necessary. I am not fully convinced that this model is better than other commonly-used sequential models.
>
> $\textbf{Response:}$ Thanks for the suggestion to compare our approach, which is a one-step method with sequential models that are capable of scaling to multi step predictions. We have compared our approach with the following sequential models: 1) Gated Graph Recurrent Neural Network (GGRNN) [1]; 2) Graph Transformer Network  (GTN) [2,3]. We find that our approach outperforms the GGRNN model by a substantial margin in both roll-out prediction and energy conservation while only having half the parameter count and requiring lesser computation time to train and performance inference. We utilize GTN to perform both one-step predictions (no temporal recurrence) by giving the entire state vector as input and predicting the acceleration targets corresponding to the current state and multi-step predictions (temporal recurrence) by predicting acceleration targets for each state. We find that our approach outperforms single step GTN in 8 out of 10 datasets while outperforming multi step GTN in 4 out of 10 datasets. While the difference in roll-out MSE are not substantial between GTN and our approach, this shows that multi-step GTN is an effective approach to learning from temporal data. However, while multi-step GTN achieves superior performance, its parameter count (approx. 40 million) is a magnitude higher than our approach (approx. 3 million), therefore having a high space complexity. Further, we find that the performance of GTN in comparison to our approach is not consistent across datasets. In some datasets, such as 2D and 3D Gravity($r^2$), we notice a significant drop in predictive performance.
>
> In addition to performing roll-out error comparison, we also present additional analysis on energy conservation across a simulation. We find that our approach outperforms GTN in 7 out of 10 datasets without experiencing any extreme energy blow-out. This shows that while GTN is able to capture temporal dependencies well, it is unable to approximate the true dynamics of the system and respect energy conservation symmetry. Hence we find that our model is light-weight, respect energy symmetries better while having almost similar predictive performance as GTN. However, due to the training time, we were unable to run the experiments for different look back lengths. We hypothesize that the roll-out MSE performance may change when compared using the optimum look back length. We also include a section on time-complexity analysis and parameter count in the supplementary material.
>
> >The idea of the autoregressive model seems not very novel.
>
> $\textbf{Response:}$ We agree that autoregressive models are prevalent across Machine learning and have led to insightful findings and improvements in various tasks. Our work is novel for the following reasons:
>
> - Our autoregressive encoder operates at the data-level as opposed to the latent space level, therefore encoding state space dependencies  in the latent space. We have included a paragraph in the Introduction that discusses and highlights the various spatial and spatio-temporal dependencies captured by a linear encoder and an autoregressive encoder. An example of such a dependency is that future states do not affect the past states while past states affect the future states.
>
> - Majority of the work [4,5,6,7] in the space of graph-based physics learning has thus far focused on improving single-step prediction efficiency. To our knowledge, [2] is the most recent work that explores a temporal model to learn entire trajectory sequences by using a Graph Transformer model. Further [2] requires pre-computing latent embeddings and well as involves sequential computations to predict the latent states during Transformer training. In comparison, our approach computes spatio-temporal latent embeddings in a parallel manner while encoding temporal and state space dependencies on the latent space.
>
> - Hence, our work is the first to propose a light weight, non-sequential temporal model that can capture temporal biases while preserving important physics constraints such as conservation of energy by training only on single-step particle physics simulations.
>
> We hope that our responses have addressed your questions. If you have any remaining suggestions or comments, we are happy to address them. Again, we thank you for your feedback and suggestions to improve our paper.

---

> > ### Author Response · Authors · 2022-11-19
> > **Response to Reviewer n2LJ**
> >
> > Dear Reviewer,
> >
> > Thanks again for your review. The discussion period is almost ending. We hope we have addressed your questions and concerns. Please let us know if there are anymore questions/concerns that we can address to make our paper better.

---

> > ### Comment · Reviewer_n2LJ · 2022-11-26
> > **Keep my score.**
> >
> > I would thank the authors' detailed response. I think this work has some merits, but it is still under the bar of acceptance considering the quality of this version and the other reviews. I recommend that the authors improve their paper according to these comments in a future revision.

---

> ### Author Response · Authors · 2022-11-17
> **References**
>
> $\textbf{References:}$ \
> [1] Youngjoo Seo, Michael Defferrard, Pierre Vandergheynst, and Xavier Bresson. Structured sequence modeling with graph convolutional recurrent networks. In International conference on neural information processing, pp. 362–373. Springer, 2018. \
> [2] Xu Han, Han Gao, Tobias Pffaf, Jian-Xun Wang, and Li-Ping Liu. Predicting Physics in Mesh-
> reduced Space with Temporal Attention. arXiv preprint arXiv:2201.09113, 2022. \
> [3] Yunsheng Shi, Zhengjie Huang, Shikun Feng, Hui Zhong, Wenjin Wang, and Yu Sun. Masked label
> prediction: Unified message passing model for semi-supervised classification. arXiv preprint
> arXiv:2009.03509, 2020. \
> [4] Sanchez-Gonzalez, A., Godwin, J., Pfaff, T., Ying, R., Leskovec, J., & Battaglia, P. (2020, November). Learning to simulate complex physics with graph networks. In International Conference on Machine Learning (pp. 8459-8468). PMLR. \
> [5] Pfaff, T., Fortunato, M., Sanchez-Gonzalez, A., & Battaglia, P. W. (2020). Learning mesh-based simulation with graph networks. arXiv preprint arXiv:2010.03409. \
> [6] Rubanova, Y., Sanchez-Gonzalez, A., Pfaff, T., & Battaglia, P. (2021). Constraint-based graph network simulator. arXiv preprint arXiv:2112.09161. \
> [7] Stachenfeld, K., Fielding, D. B., Kochkov, D., Cranmer, M., Pfaff, T., Godwin, J., ... & Sanchez-Gonzalez, A. (2021). Learned Coarse Models for Efficient Turbulence Simulation. arXiv preprint arXiv:2112.15275. \

---

### Official Review · Reviewer_hzQp · 2022-10-25

**Confidence:** 2
**Correctness:** 3
**Technical Novelty And Significance:** 2
**Empirical Novelty And Significance:** 2
**Recommendation:** 5

**Clarity, Quality, Novelty And Reproducibility:**

The description of the approach in Section 3 was a bit difficult to parse. This section could be better developed so that the reader can appreciate the important aspects of the approach.
- In section 3.1, the dimensions of $\boldsymbol{r}$ should be $N \times d$ instead of $Nxd$ ?
- In the text following Eq 2, it should be Eq 2 instead of Eq 1 ?
- I presume $act$ is an activation function. It would be helpful to clearly state so.
- Terms like $H(x)$ are not clearly defined. It would be useful to clearly define what output it corresponds to.
- The definition of the quantity $m(s,t)$ is hard to parse, and it is not immediately clear what $m$, $s$, $n$, $p$ stand for
- The input to the MLP edge encoder already contains the acceleration terms, is this correct?

It would also be useful to have a legend or more detailed captions for figures 7 and 8. It looks like the colored traces are the ground truth trajectories, and the gray traces are the estimates?

There are also several typos and grammatical errors.

**Strength And Weaknesses:**

The experimental results demonstrate that the AGN outperforms the baseline GNs on most datasets presented.

However, some aspects of the approach were not completely clear.
- In the introduction, the authors mention that the encoder learns the arrow of time. But based on the discussion following Eq 2, it looks like the mask M explicitly induces the desired temporal dependencies?
- In the last paragraph of section 3.2.1 (VAR encoder), the input to the VAR edge encoder includes the derivatives of the full state: $\dot{\boldsymbol{x}_i}(t) - \dot{\boldsymbol{x}_j}(t)$. According to the definition following Eq 1, these derivative terms already include the target $\ddot{\boldsymbol{r}}$. Is this a typo or intended? Also, why does Eq 5 include both $H_i - H_j$ and $|H_i - H_j|$ terms?
- If the temporal dependence is already captured by the embeddings, why are there $L$ message-passing steps? How much more computationally efficient is the AGN compared to the baseline GNs used in the experiments?

**Summary Of The Paper:**

This work presents an Autoregressive Graph Network (AGN) that learns physics models. Given the state of a system for $k$ time-steps, an autoencoder is used to generate node (and edge) embeddings for each particle (and pairs of particles) in the system. Importantly, the autoencoder is constrained to impose an autoregressive structure that captures temporal dependencies. The node and edge embeddings are then input to a message-passing network, whose output is then used to predict the target state.
Finally, experiments are performed to study the performance of the AGN at multi-step forward prediction on datasets featuring different particle interactions.

**Summary Of The Review:**

This paper presents a Graph Network that imposes an autoregressive structure on its input embeddings to capture temporal dependencies.  Experimental results indicate that the approach outperforms baseline GNs on multiple datasets. However, a few important aspects of the approach were unclear. Improving the presentation, for section 3 in particular, would help the reader better understand these aspects of the approach.

---

> ### Author Response · Authors · 2022-11-17
> **Response to Reviewer hzQp**
>
> We sincerely thank you for your time and constructive feedback. We address your comments and questions below:
> >In the introduction, the authors mention that the encoder learns the arrow of time. But based on the discussion following Eq 2, it looks like the mask M explicitly induces the desired temporal dependencies?
>
> $\textbf{Response:}$ We agree that our encoder is explicitly induced with the desired temporal dependencies by masking out connections between past states and future states. We have modified some of our statements in the paper to reflect just the same. In addition, we have provided concrete motivation in the Introduction for why it is important to capture such temporal dependencies and what type of dependencies are captured by a linear encoder. In summary, linear encoders learn dependencies between past and future states, therefore learning temporal biases such as the future states affect the past states. However, in practice, neural networks do not encounter examples wherein the future affects the past states, therefore learning biases that may not hold during testing. Hence, by explicitly providing the causal influence of the past states on the future states, we observe that our model is able to predict trajectories longer than they have seen during training while conserving energy accumulation as well as roll-out error accumulation.
>
> >In the last paragraph of section 3.2.1 (VAR encoder), the input to the VAR edge encoder includes the derivatives of the full state: $\dot{\textbf{x}}_i(t) - \dot{\textbf{x}}_j(t)$. According to the definition following Eq 1, these derivative terms already include the target . Is this a typo or intended? Also, why does Eq 5 include both $H_i - H_j$ and $|H_i - H_j|$ terms?
>
> $\textbf{Response:}$ No it is not a typo, in our dataset, velocity and acceleration are instantaneous variables. Here, $\dot{\textbf{x}}_i(t) - \dot{\textbf{x}}_j(t)$ simply denotes the relative velocity between particles and not how velocity changes over time for a particle. Further, we include both $H_i - H_j$ and $|H_i - H_j|$ terms following [1]. While the authors only compute the relative distances between particles and the corresponding norm as inputs to the encoder, we maintain the same edge feature computation through the message passing steps. This essentially corresponds to performing edge convolutions [2] using particle specific latent features while also concatenating the edge features from the previous state and the norm of  $H_i - H_j$. We noticed a slightly better performance across all models by performing such as form of edge convolution.
>
> > If the temporal dependence is already captured by the embeddings, why are there $L$ message-passing steps? How much more computationally efficient is the AGN compared to the baseline GNs used in the experiments?
>
> $\textbf{Response:}$ While performing a coarse hyper-parameter optimization, we found that increasing message passing steps lead to better prediction accuracy. For a simple encode-process-decode Graph Network constructed using MLPs, we find that there wasn't any performance gains by increasing beyond 3. However, in the case of a Graph Transformer Network, we found that just a single message passing step is sufficient.
>
> During this rebuttal period, we have added additional experiments to our paper wherein we compare the autoregressive approach with sequential models such as Gated Graph Recurrent Neural Network and Graph Transformer Network. We find that our method performs better than the RNN while performing on par with Graph Transformer Network. Further, while the sequential models are trained to predict an acceleration target for each state in a sample trajectory of length k, our method is trained to perform only one-step predictions. In addition, we have included a section dedicated to time complexity analysis between the baselines as well as model parameter count. We note that our model has an order of magnitude fewer number of parameters (approx. 3 million) in comparison to the sequential models (approx. 40 million in the case of Transformer).
>
> [1] Sanchez-Gonzalez, A., Godwin, J., Pfaff, T., Ying, R., Leskovec, J., & Battaglia, P. (2020, November). Learning to simulate complex physics with graph networks. In International Conference on Machine Learning (pp. 8459-8468). PMLR.
>
> [2] Wang, Y., Sun, Y., Liu, Z., Sarma, S. E., Bronstein, M. M., & Solomon, J. M. (2019). Dynamic graph cnn for learning on point clouds. Acm Transactions On Graphics (tog), 38(5), 1-12.

---

> > ### Author Response · Authors · 2022-11-19
> > **Response to Reviewer hzQp**
> >
> > Dear Reviewer,
> >
> > Thanks again for your review. The discussion period is almost ending. We hope we have addressed your questions and concerns. Please let us know if there are anymore questions/concerns that we can address to make our paper better.

---

> > > ### Comment · Reviewer_hzQp · 2022-12-02
> > > **Thank you for the response**
> > >
> > > I appreciate your detailed response and for incorporating the suggestions in your edits. The work is promising, however, I still think the paper needs to be a bit more substantially rewritten before it is ready for publication.

---

> ### Author Response · Authors · 2022-11-17
> **Response to Reviewer hzQp**
>
> > In section 3.1, the dimensions of  should be  instead of  ?
> In the text following Eq 2, it should be Eq 2 instead of Eq 1 ?
> I presume  is an activation function. It would be helpful to clearly state so.
> Terms like  are not clearly defined. It would be useful to clearly define what output it corresponds to.
> The definition of the quantity  is hard to parse, and it is not immediately clear what , , ,  stand for
> The input to the MLP edge encoder already contains the acceleration terms, is this correct?
>
> $\textbf{Response:}$ We thank you for pointing out the inconsistencies in our paper. We have made the corresponding changes to our revision and have enhanced clarity on the explanation of the function $m(s,t)$ and in general also enhanced the presentation of the paper. We confirm again that acceleration terms were not provided as input to any of the models
>
> We hope that our responses have addressed your questions. If you have any remaining suggestions or comments, we are happy to address them. Again, we thank you for your feedback and suggestions to improve our paper.

---

### Decision · Program_Chairs · 2023-01-20

**Decision:**

Reject

**Justification For Why Not Higher Score:**

- The novelty and clarity of the method should be improved
- A more thorough experimental evaluation is needed

**Justification For Why Not Lower Score:**

N/A

**Metareview: Summary, Strengths And Weaknesses:**

This paper introduces an  Autoregressive Graph Network (AGN) for particle-based prediction in physical problems. The idea is to augment standard graph neural networks (GNN) with an autoregressive structure in the temporal domain. The model is shown to perform on par with Graph Transformers while being much more efficient.
The paper initially received two reject (3) recommendations and two borderline reject (5) recommendations. The main concerns pointed out by the reviewers related to clarifications in the method and contributions, and requested consolidated experiments including additional baselines for comparison. The answers provided by the authors during the discussion period were not sufficient to convince the reviewers to increase their grades, and the reviewers considered that the paper should be re-written and re-submitted. After rebuttal, there remained a consensus that the paper should be rejected.

The AC carefully reads the submission and discussions. The AC considers that the description and positioning of the paper should be improved and clarified, especially to consolidate the originality of the approach beyond combining auto-regressive models and GNNs, which seems too limited. The experiments should also include more comparisons to standard and stronger baselines. Therefore, the AC recommends rejection.